# *Aeromonas* spp. Prevalence, Virulence, and Antimicrobial Resistance in an *Ex Situ* Program for Threatened Freshwater Fish—A Pilot Study with Protective Measures

**DOI:** 10.3390/ani12040436

**Published:** 2022-02-11

**Authors:** Miguel L. Grilo, Guadalupe Amaro, Lélia Chambel, Carolina S. Marques, Tiago A. Marques, Fátima Gil, Carla Sousa-Santos, Joana I. Robalo, Manuela Oliveira

**Affiliations:** 1CIISA—Centre for Interdisciplinary Research in Animal Health, Faculty of Veterinary Medicine, University of Lisbon, 1300-477 Lisbon, Portugal; lusowesterosi@gmail.com; 2MARE—Marine and Environmental Sciences Centre, ISPA—Instituto Universitário de Ciências Psicológicas, Sociais e da Vida, 1149-041 Lisbon, Portugal; csousasantos@gmail.com (C.S.-S.); jrobalo@ispa.pt (J.I.R.); 3BioISI—Biosystems and Integrative Sciences Institute, Faculdade de Ciências, Universidade de Lisboa, 1749-016 Lisbon, Portugal; lmchambel@fc.ul.pt; 4Departamento de Biologia Animal, Centro de Estatística e Aplicações, Universidade de Lisboa, 1749-016 Lisbon, Portugal; carolinasegmarques@gmail.com (C.S.M.); tiago.marques@st-andrews.ac.uk (T.A.M.); 5Centre for Research into Ecological & Environmental Modelling, University of St Andrews, St Andrews KY16 9LZ, UK; 6Aquário Vasco da Gama, 1495-718 Cruz Quebrada-Dafundo, Portugal; avg.aqua@marinha.pt

**Keywords:** *ex situ*, *Aeromonas* spp., *Iberochondrostoma lusitanicum*, virulence, antimicrobial resistance

## Abstract

**Simple Summary:**

Knowledge regarding best practices to prevent bacterial disease and antimicrobial resistance acquisition in aquatic *ex situ* programs is limited. This pilot study aimed to investigate the role of protective measures in the prevalence, antimicrobial resistance profiles, and virulence signatures of *Aeromonas* spp. in Portuguese nase (Iberochondrostoma lusitanicum) kept in an *ex situ* program. Fish were randomly divided into two tanks (i.e., with and without protective measures). Bacterial sampling was performed weekly for 5 weeks, and *Aeromonas* spp. prevalence, antimicrobial resistance, and virulence signatures were compared. We observed an increase in antimicrobial resistance among collected isolates over the experiment duration, with a trend of *Aeromonas* spp. prevalence and virulence decreasing when using protective measures. This pilot study sheds light on *Aeromonas* spp. prevalence, antimicrobial resistance, and virulence dynamics in aquatic *ex situ* programs, while constituting a first approach in the determination of the potential use of protective measures in such settings.

**Abstract:**

*Ex situ* breeding programs are important conservation tools for endangered freshwater fish. However, developing husbandry techniques that decrease the likelihood of disease, antimicrobial resistance, and virulence determinants acquisition during this process is challenging. In this pilot study, we conducted a captivity experiment with Portuguese nase (*Iberochondrostoma lusitanicum*), a critically endangered leuciscid species, to investigate the influence of simple protective measures (i.e., material disinfection protocols and animal handling with gloves) on the dynamics of a potential pathogenic genus, *Aeromonas*, as well as its virulence profiles and antimicrobial resistance signatures. Our findings show that antimicrobial resistance in *Aeromonas* spp. collected from *I. lusitanicum* significantly increased during the extent of the assay (5 weeks), with all isolates collected at the end of the study classified as multidrug-resistant. Additionally, humans handling fishes without protective measures were colonized by *Aeromonas* spp. The use of protective measures suggested a decreasing trend in *Aeromonas* spp. prevalence in *I. lusitanicum*, while bacterial isolates displayed significantly lower virulence index values when virulence phenotypical expression was tested at 22 °C. Despite this study representing an initial trial, which needs support from further research, protective measures tested are considered a simple tool to be applied in *ex situ* breeding programs for aquatic animals worldwide. Furthermore, current results raise concern regarding antimicrobial resistance amplification and zoonotic transmission of *Aeromonas* spp. in aquatic *ex situ* programs.

## 1. Introduction

*Ex situ* breeding and recovery programs are important conservation tools that help to secure species experiencing severe declines in their natural habitat [1]. In Portugal, a dedicated *ex situ* breeding program was established in 2008 aiming to counteract the threatened status of endemic leuciscid species [2]. This group of freshwater fishes, presenting a high level of endemicity in the Iberian Peninsula, faces severe conservation constraints associated with habitat degradation, water extraction, summer droughts, and the proliferation of invasive species [2]. Despite the conservational value of *ex situ* programs for wild species, these actions are not met without a cost. The translocation of species from their original habitat into anthropogenic facilities, along with the changes implemented in the husbandry of the individuals during their stay in the program and inherent changes in phenotypical and genotypical traits of the animals caused by captivity, are accompanied by important modifications in the animal’s microbiota [3]. Changes in the host microbiota composition and traits can impair overall fitness and present severe consequences for the individual survival [4].

Additionally, and since dissemination of antimicrobial resistance and virulence determinants between different environments are a general public health concern, the role of recovery and breeding programs as gateways of antimicrobial resistance and virulence transfer between anthropogenic cycles and natural environments needs to be addressed in the One Health context. Previous studies have addressed the effect that recovery programs have had in the acquisition of resistance determinants by wild animals in close contact with humans [5,6]. However, no study so far has explored such dynamics in *ex situ* conservation programs with aquatic species. The relocation of these animals into their natural habitats may establish new communication bridges and the formation of additional resistance and virulence determinants reservoirs in natural environments, which are difficult to be controlled and eradicated.

It is fundamental to understand current reality and adapt captivity’s conditions and husbandry techniques in order to minimize alterations in the host microbiota during the program extent, as well as to prevent the acquisition and further dissemination of resistance and virulence determinants that can constitute reservoirs when released into the wild.

In this pilot study, in order to understand the dynamics of prevalence and antimicrobial resistance and virulence determinants of *Aeromonas* spp., an important zoonotic and fish pathogenic agent, we developed a captivity experiment with Portuguese nase (*Iberochondrostoma lusitanicum*) individuals under two husbandry regimens with different biosafety measures and evaluated the prevalence and structure of *Aeromonas* spp., their antimicrobial resistance signatures, and virulence profiles.

## 2. Materials and Methods

### 2.1. I. lusitanicum Capture and Transport

The *ex situ* conservation program, ongoing at the Vasco da Gama Aquarium (Lisbon, Portugal), is responsible for the captive breeding, for restocking purposes, of five threatened leuciscids (*Achondrostoma occidentale*, *Anaecypris hispanica*, *Squalius pyrenaicus*, *Iberochondrostoma almacai,* and *Iberochondrostoma lusitanicum*). Populations of each species considered to be at a higher risk are selected to be included in the program and a stock of wild adults is collected from the natural habitat and housed in separate tanks to prevent contact.

Due to space limitation in the program, selection of populations for breeding is performed through a rotation scheme. This study benefited from the fact that a new wild stock of *I. lusitanicum* from the Sado river was going to be established at the Vasco da Gama Aquarium facilities. Hence, this species was selected to participate in this study.

Fishing and detention licenses were granted by the competent authority—Instituto da Conservação da Natureza e das Florestas, IP (Fishing Credential nº 71-A/2018; License nº 438/18/CAPT). The animals included in this study were cared for according to the rules given by the current EU (Directive 2010/63/EC) and national (DL 113/2013) legislation and by the competent Portuguese authority (Direção Geral de Alimentação e Veterinária, DGAV, www.dgv.min-agricultura.pt/portal/page/portal/DGV, accessed on 22 June 2021). In the scope of this study, only non-invasive samples were collected during the routine procedures, and no ethics committee approval was needed. Samples were obtained by trained veterinarians, following standard routine procedures. No animal experiment has been performed in the scope of this study.

Animals were collected by electrofishing following standard procedures [7]. Captured fish were transferred to a container with water collected at the sampling site in order to monitor their stunning recuperation. The animals included in this study were collected at Ribeira de Grândola (38.168980°, −8.569030°) in May, 2018. In total, 22 adult individuals were collected (eight females, six males, and eight undetermined; mean size = 73.9 mm ± 10.3 SD).

After capture, animals were placed in a transport container with water collected from the river stream. The container dimensions were appropriate for the expected animal abundance in order to guarantee the absence of casualties during transportation (50 L for 30 individuals). Animals were transported by car to Vasco da Gama Aquarium.

### 2.2. Experimental Setup

In order to mimic the breeding program conditions, the experiment was set in the same area where the breeding tanks are usually located—an exterior terrace, under natural light, temperature, and pluviosity at the Vasco da Gama Aquarium. Experimental conditions applied during this study are summarized in Table 1.

The experimental setting consisted of two 100 L tanks with closed water circulation and sand filtration systems (obtained from tanks already established). Two fish refuges, made with PVC tubes, were included in each tank. The tanks were disinfected with 70% ethanol prior to their filling with water. Circulation, filtration, and refuge materials were immersed in 70% ethanol for 24 h, followed by periods of 15 min exposure to UV radiation of all their surfaces (the number of periods varied according to the number of surfaces of the object). Replicates of each condition (i.e., tanks) were not performed due to technical limitations (i.e., tank uniformity and availability) and sample size (i.e., low number of animals, gregarious behavior of *I. lusitanicum*, stress effects due to group separation and possible interference with reproduction, high microbiota inter-variability among individuals).

*Aeromonas* spp. screening was performed from the following samples: swabs from the tanks’ walls and floor after disinfection, food (frozen mosquito larvae, frozen Mysis shrimp, and frozen krill), water samples from the aquarium supplier (directly from the tap, 36 h prior to tank filling), and filtering sand.

Tank preparation was initiated 120 h prior to fish introduction to allow the water to stabilize and to prevent shock, leading to fish mortality. Tank conditions differed regarding contact with the environment. In the control tank, a fine mesh (ca. 2 mm) plastic cover was used to cover the top of the tank, still allowing the entrance of external agents (i.e., avian feces, insects, rain-water, dust), as happens in the tanks currently used in the program. In the test tank, a plastic cover was used over the surface of the tank, restraining the entrance of external agents.

The feeding regimen followed the protocol established for this species in captivity [2]. Food sources were limited to frozen mosquito larvae, Mysis shrimp, and krill. Food was thawed two hours prior to feeding and cleaned with water to eliminate impurities. Food for animals at the control tank was cleaned with water directly from tap and stored in current use containers (not exposed to disinfection protocols), while food for the animals at the test tank was cleaned with sterile water stored at sterile shots and stored in containers disinfected with 70% ethanol and exposed to UV radiation.

Water renovation was performed weekly and consisted of the extraction of 10 L of tank water and the slow introduction of the same amount of tap water.

Handling of animals, food, and water from the control tank was performed by two rotating aquarists not subjected to disinfection protocols. Any action directed towards the test tank was performed by an external specialized operator, using nitrile gloves disinfected with 70% ethanol. Bacterial sampling was performed weekly on a fasting day. Sampling materials used in the test tank (shrimp net and water suction system) were exposed to UV radiation. Materials used in the control tank were not disinfected.

By the end of the assay, animals were introduced into the breeding tanks of the *ex situ* conservation program.

### 2.3. Assay with Distinct Husbandry Protocols

A weekly scheme of the assay, including tasks performed on each day, is displayed in Appendix A. The assay was performed for 5 weeks.

Following arrival to Vasco da Gama Aquarium, the animals were randomly selected and divided into two groups (*n* =11, each; test group—mean size =72.5 mm ± 10 SD, control group—mean size =75.3 mm ± 10.4 SD). Each animal was measured, sex was determined (when possible), and a body swab (cloacal area, lateral sides of the body and fins, excluding head; ESwab LiquidAmies Collection and Transport System, ThermoFisher Scientific, Waltham, MA, USA) was collected to establish an initial bacteriological baseline. Afterwards, animals were introduced to the corresponding tank.

In each bacteriological sampling action, water from each tank was transferred to a separate tank, followed by the transfer of the individuals collected with shrimp nets. Animals were individually measured, a swab sample was performed, and the animals were returned to their respective tanks. Different measuring devices and handling tanks were used between individuals from different tanks to prevent cross-contamination (Appendix A).

In the control tank, the aquarist did not use gloves or disinfect their hands before handling the animals. From the 3^rd^ to the 5^th^ bacteriological sampling action in the control tank, both prior to and after the procedures, a swab was collected from the hands, fingernails, and lower arms of the aquarist. Human sampling was performed after the individuals were informed regarding the sampling procedure, and signed an informed consent in accordance with the Helsinki Declaration of the World Medical Association (version October 2013) and the Oviedo Convention (version April 1997). No human experiment has been performed in the scope of this study.

Food and water samples were collected weekly, at regular days, by using sterile 50 mL tubes (Corning Life Sciences, New York, NY, USA).

Water quality was controlled over the experiment’s period by determination of a set of physical and chemical parameters: pH (mean = 7.02 ± 0.5 SD) and temperature (mean = 14.9 °C ± 0.8 SD), using a portable waterproof pH meter model HI98130 (Hanna Instruments^®^, Woonsocket, RI, USA); nitrites (mean = 0 mg/L ± 0 SD) and nitrates (mean = 0 mg/L ± 0 SD), using colorimetric strips (ITS Thorsten BetzelTM, Hattersheim, Germany).

### 2.4. *Aeromonas* spp. Isolation, Genomic Typing, and Identification

Selection of *Aeromonas* spp. as model pathogens was based on their relevance as fish pathogens. *Aeromonas* spp. are one of the most common pathogen groups of freshwater fishes, and although their impact in Iberian leuciscid populations is unknown, their impact in both wild and farmed fishes is emphasized, especially in stressful conditions such as the translocation into new environments [8].

After collection, swabs (from human, animal, and food samples) were inoculated in tubes with 8 mL of brain heart infusion (BHI) broth (VWR, Radnor, PA, USA), subjected to homogenization, and incubated at 37 °C for 24 h. Regarding water samples, tubes containing the samples were homogenized briefly, and 100 µL of each sample were transferred into BHI broth, followed by a similar protocol to the one applied to the rest of the samples. Following incubation, a sample from each tube was transferred to glutamate starch red phenol (GSP) agar plates supplemented with 100,000 IU sodium penicillin g/L (Merck, Kenilworth, NJ, USA). Plates were incubated at 37 °C for 12 h. Four distinct colonies displaying *Aeromonas* spp. morphology were randomly selected from each individual sample and isolated into pure cultures on BHI agar, at 37 °C for 24 h. *Aeromonas hydrophila* ATCC 7966 was used as a positive control.

Isolates were characterized regarding Gram-staining and oxidase activity. Pure cultures of oxidase positive, Gram-negative rods were stored in buffered peptone water (VWR, Radnor, PA, USA) with 20% glycerol at −80 °C during the study.

Bacterial genomic DNA was obtained by the boiling method, after growth on BHI agar (37 °C, 24 h), as described before [9].

In order to perform the molecular typing of the isolates, a random amplified polymorphic DNA (RAPD) technique was used. The method was applied as described before [10,11], with minor modifications. Primers Ap3 and Ap5 [10] were chosen and used in independent mixtures to achieve fingerprinting patterns of the isolates. Each amplification reaction was performed in a final volume of 25 µL. The mixture consisted of: 12.5 µL of Supreme NZYTaq 2× Green Master Mix (NZYTech, Lisbon, Portugal), 8.5 µL of PCR-grade water (Sigma-Aldrich, St. Louis, MO, USA), 2.5 µL of bovine serum albumin (0.01%; Thermo Fisher Scientific, Waltham, MA, USA), 0.5 µL (1 µM) of primer, and 1 µL of template DNA. Thermocycler conditions used included 94 °C for 5 min; 40 cycles of 94 °C for 45 s, 40 °C for 1 min, and 72 °C for 2 min; 72 °C for 5 min.

PCR products were resolved by gel electrophoresis (1.5% (*w*/*v*) agarose in 1× TBE Buffer (NZYTech, Lisbon, Portugal)) for 50 min at 90 V. As a molecular weight marker, NZYDNA Ladder VII (NZYTech, Lisbon, Portugal) was used. The visualization of gels was performed using a UV light transilluminator. Images were captured using the Bio-Rad ChemiDoc XRS imaging system (Bio-Rad Laboratories, Hercules, CA, USA).

In order to achieve species identification, a multiplex PCR protocol, previously described by Persson et al. [12], was used with minor modifications. This protocol discriminates between *A. caviae*, *A. media*, *A. hydrophila,* and *A. veronii*, and contains an internal control for the genus *Aeromonas*. PCR mixtures were performed in a final volume of 25 µL and were composed of 12.15 µL of Supreme NZYTaq 2× Green Master Mix (NZYTech, Lisbon, Portugal), 10 µL of PCR-grade water (Sigma-Aldrich, St. Louis, MO, USA), 0.025 µL (0.05 µM) of primers A-16S, 0.25 µL (0.5 µM) of primers A-cav, 0.1 µL (0.2 µM) of primers A-med, 0.225 µL (0.45 µM) of primers A-hyd, 0.075 µL (0.15 µM) of primers A-Ver, and 1.5 µL of template DNA. Thermocycler (VWR, Radnor, PA, USA) conditions were as follows: 95 °C for 2 min, followed by 6 cycles of 94 °C for 40 s, 68 °C for 50 s, and 72 °C for 40 s; 30 cycles at 94 °C for 40 s, 66 °C for 50 s, and 72 °C for 40 s. *A. caviae* ATCC 1976, *A. hydrophila* ATCC 7966, *A. media* ATCC 33907, and *A. veronii* ATCC 35624 were used as positive controls.

Amplification products were resolved by gel electrophoresis and visualized as described above. Gels were resolved for 45 min at 90 V. As a molecular weight marker, NZYDNA Ladder VI (NZYTech, Lisbon, Portugal) was used.

### 2.5. Virulence Factors Screening

The virulence factor expression by the isolates was accessed by a set of phenotypical assays, as established in protocols previously described, with minor modifications. Screening was performed on isolates collected during the first and the fifth sampling week. The following virulence factors were investigated, namely: (1) gelatinolytic activity, using Oxoid nutrient gelatin (Thermo Fisher Scientific, Waltham, MA, USA) for 24 h [13]; (2) hemolytic activity, using Columbia agar supplemented with 5% sheep blood (VWR, Radnor, PA, USA) for 24 h [14]; (3) lipolytic activity, using spirit blue agar (Difco, Franklin Lakes, NJ, USA) supplemented with 0.2% Tween 80 (VWR, Radnor, PA, USA) and 20% olive oil (commercial) for 8 h [15]; (4) proteolytic activity, using skim milk agar (Sigma-Aldrich, St. Louis, MO, USA) for 24 h [16]; (5) slime production, using Congo red agar (VWR, Radnor, PA, USA) for 72 h [17]. Two incubation temperatures were used. The 22 °C was an average based on river’s water temperature data collected during annual monitoring census performed in the summer seasons of 2017 to 2019 (Sousa-Santos, personal communication). This was performed in order to mimic fish’s body temperature since they are poikilothermic. The 37 °C was used to mimic human body temperature.

The following strains were used as controls: *A. caviae* ATCC 15468 (hemolysin negative), *A. hydrophila* ATCC 7966 (hemolysin positive), *Enterococcus faecium* EZ40 clinical isolate from canine periodontal disease (slime producer), *Escherichia coli* ATCC 25922 (gelatinase negative; slime non-producer), *Pseudomonas aeruginosa* Z25.1 clinical isolate diabetic foot infection (protease and gelatinase positive; lipase negative), and *Staphylococcus aureus* ATCC 29213 (lipase positive, protease negative). *P. aeruginosa* and *E. faecium* [18,19] belong to the bacterial collection of the Laboratory of Microbiology and Immunology, Faculty of Veterinary Medicine, University of Lisbon, Portugal.

The virulence index of each isolate was calculated based on the ratio between positive tests for virulence factors and the total amount of virulence factors tested [20].

### 2.6. Antimicrobial Susceptibility Testing

Antimicrobial susceptibility testing was performed using the disk diffusion technique [21]. Guidelines and breakpoints of the Clinical and Laboratory Standards Institute were followed as reference [22,23]. The following antibiotics (Mastdiscs, Mast Group, Liverpool, United Kingdom) were tested: amikacin (AK, 30 µg), amoxicillin/clavulanic acid (AUG, 20–10 µg), aztreonam (ATM, 30 µg), ceftazidime (CAZ, 30 µg), enrofloxacin (ENF, 5 µg), erythromycin (E, 15 µg), florfenicol (FFC, 30 µg), imipenem (IMI, 10 µg), nitrofurantoin (NI, 300 µg), streptomycin (S, 10 µg), tetracycline (T, 30 µg), and sulfamethoxazole/trimethoprim (TS, 23.75–1.25 µg). Antimicrobial compound choice followed those commonly used to treat Gram-negative infections in human and veterinary medicine, as well as those compounds used for treating aquatic animals’ diseases. *Escherichia coli* ATCC 25922 was used as a quality control.

Isolates were categorized as multidrug-resistant, as described by Magiorakos et al. [24], when presenting non-susceptibility to at least one antimicrobial compound in three or more antimicrobial categories. Multiple antibiotic resistance (MAR) index values were produced for each isolate and calculated based on the ratio between the number of antimicrobial compounds to each the isolate presenting a non-susceptibility profile and the total amount of antimicrobial compounds tested [25]. Non-susceptibility was defined as presenting intermediate or resistant category status. Antimicrobial compounds to which *Aeromonas* spp. are considered intrinsically resistant (amoxicillin/clavulanic, erythromycin and streptomycin) were not included in the multidrug resistance characterization and in the MAR index calculation.

### 2.7. Statistical Analysis

In order to analyze the reproducibility level of the molecular species identification, phenotypic virulence expression, antimicrobial susceptibility testing, and genomic typing techniques, a random sample including 10% replicates was used.

BioNumerics version 7.6 software (Applied Maths, Sint-Martens-Latem, Belgium) was used to perform genomic typing. The similarity of the fingerprinting patterns was achieved based on a dendrogram calculated with the Pearson correlation coefficient. Cluster analysis was performed through the unweighted pair group method with arithmetic average (UPGMA). The reproducibility value of the technique was determined as the average similarity value of all replicates’ pairs (91.88%). When patterns presented higher similarity values, they were considered to be undistinguishable. Clusters were formed based on a joint evaluation of the fingerprinting profiles and the *Aeromonas* species. For all dendrograms and all clusters, the lowest similarity value was investigated and a cut-off value was established across dendrograms to form clusters and enable comparisons between dendrograms.

Several isolate level response variables were modeled as a function of tank and sampling week. At the isolate level, and therefore, using a GLMM (generalized linear mixed-effects model [26]) with fish as a random effect, we modeled, as a binomial logistic response, the virulence factors (i.e., prevalence of individual activity of each of the tested virulence factors) at both (1) 22 °C and (2) 37 °C (0-Negative, 1-Positive), and the (3) categories of susceptibility to antibiotics (0—Non-susceptible, 1—Susceptible), as well as, using a beta response (continuous values ranging from 0 to 1), the (4) MAR index values (i.e., the ratio between the number of antimicrobial compounds considered as presenting acquired non-susceptibility and the total amount of antimicrobial compounds tested) and the virulence index values (i.e., the ratio between the number of virulence factors for which positive activity was detected and the total amount of virulence factors tested) at both (5) 22 °C and (6) 37 °C.

Using a multinomial log-linear model (package nnet, version 7.3-15) [27], the (7) proportion of the different species of *Aeromonas* was considered.

Considering an analysis at the tank level, a GLM with a beta response was used to model the (8) Simpson index and the (9) prevalence of *Aeromonas* spp. The statistical analysis was done using R software [28]. Graphs were produced using GraphPad Prism (GraphPad Software, San Diego, CA, USA, version 5.01).

## 3. Results

Mortality was not observed in both tanks during the extent of the assay. While individuals displayed a shy behavior in the first two days of the experiment and remained in the refuges provided, swimming, exploration, and feeding behavior was considered normal for the remaining period.

*Aeromonas* spp. isolation was achieved for all animals in the beginning of the trial. However, prevalence in each tank varied across the assay’s weeks. While *Aeromonas* spp. prevalence in the control tank was of 100% or close to it in every week, prevalence in the test tank decreased to 54.5% in the last week of sampling, although it displayed similar prevalence values to the control tank in the previous weeks (Figure 1). This variation was significantly different among tanks (*p* < 0.001).

Prior to the beginning of the captivity assay, *Aeromonas* spp. were not detected in the evaluated matrixes (food, water, and filtering sand). Similarly, no *Aeromonas* spp. were isolated from the water samples collected in both tanks across the assay. Regarding food sources, it was possible to detect *Aeromonas* spp. from the food administered in the control tank in the first and second week of sampling. No *Aeromonas* spp. were isolated from the food available in the test tank. Regarding the samples collected from aquarists’ hands and arms, all swabs obtained after handling the animals were positive for *Aeromonas* spp. However, none of the swabs collected prior to handling were positive for *Aeromonas* spp.

Initial *Aeromonas* species structures were similar between individuals from both tanks (Figure 2). Some shifts were observed along the assay weeks regarding species prevalence between the two tanks. Although *A. veronii* isolation in the test tank was possible across almost all sampling actions, such phenomenon was not observed in the control tank. Instead, an increase of *A. media* prevalence was found from the first sampling action to the second, with presence also recorded in the last week of sampling. Interestingly, the original predominance of *A. veronii* in both tanks was gradually substituted by a predominance of *A. hydrophila* in both tanks. Both the sampling tank and week significantly influenced the *Aeromonas* species’ structure. *A. veronii* was more prevalent in the test tank (*p* = 0.006) in comparison with the control tank, while an opposite trend was observed for *A. media* (*p* = 0.050). *A. veronii* prevalence also significantly shifted from week one, decreasing in the second and the third weeks (*p* < 0.001), and increasing in the final week (*p* < 0.001).

Typing the 363 isolates and evaluating their relationships based on dendrogram analysis revealed the presence of 54 clusters and 40 single-member clusters. A cut-off level of 62.93% was assumed based on the evaluation of fingerprinting patterns, *Aeromonas* species, tank of origin, and sampling week (Appendix A). In both tanks, a total of 58 clones were identified at 91.88% similarity (reproducibility level). Around half of the clones detected were collected from the same individual, while the other half were collected from different individuals in the same tank and in the same sampling week. None of the clones were isolates collected from fishes in different tanks or across sampling weeks. The only observed exceptions were the food sources and the aquarists. All six isolates of *Aeromonas* spp. collected from the food were obtained from the first and the second week of sampling and were clones. Regarding the isolates collected from the aquarists’ hands, two isolates from the same aquarist in the third and fourth sampling week were clones.

Although no specific pattern was observed for both tanks, some clusters were exclusive to a particular sampling week (Appendix A). Some clones were observed, namely clones of *A. veronii,* isolated from the food sources, clones of *A. hydrophila* in the fourth week of sampling in the control tank, clones of *A. hydrophila* in the fourth week of sampling in the test tank, and clones of *A. hydrophila* in the fifth week of sampling in the test tank. Regarding the isolates collected in the same tank and in the same sampling week (Appendix A), when strains of different *Aeromonas* species were detected, they mostly formed distinct clusters.

When testing the virulence factors’ phenotypic expression of the isolates at 22 °C (Figure 3), the virulence index values obtained for the isolates in the test tank in the fifth week were significantly lower than the ones in the control tank and the ones recorded for the first sampling week (*p* = 0.003). No differences were observed between the virulence indexes from isolates from both tanks (*p* = 0.337) at each sampling week (*p* = 0.580) when tested at 37 °C.

The prevalence of virulence factors in isolates tested at 22 °C differed among experimental conditions and sampling weeks (Figure 4). Regarding hemolytic activity, prevalence was significantly lower in the isolates from the fifth week in comparison with the first one (*p* = 0.050), with a decrease in the isolates from the control tank. Lipolytic activity was not influenced by the experimental condition (*p* = 0.984) and sampling week (*p* = 0.974). Gelatinolytic activity differed among the weeks (*p* = 0.040), being characterized by an increase in the isolates from the control tank and a decrease in the isolates from the test tank. Proteolytic activity was also influenced by the tank (*p* = 0.021) and the sampling week (*p* = 0.032), being significantly lower in the isolates from the test tank in the fifth week. Slime production was significantly different between experimental conditions (*p* = 0.050), being lower in the isolates from the test tank. Regarding the prevalence of virulence factors tested at 37 °C, the majority of the phenotypic traits did not present differences between the studied variables. The only exception was for the proteolytic activity, which was significantly different between weeks (*p* = 0.039), being lower in the isolates from the fifth week in the control tank.

The vast majority of the isolates collected in both tanks were multidrug resistant (79%). Prevalence of multidrug resistant isolates was higher in the control tank (84%) than in the test tank (74.4%). The rate of multidrug resistance varied across the sampling weeks, but all the isolates collected in the fifth week in both tanks were multidrug resistant. The MAR index values did not differ between experimental conditions (*p* = 0.911). However, a variance across the sampling weeks was observed in the fourth week (*p* < 0.001) and in the fifth week (*p* = 0.003) and MAR index values significantly increased from the first to the fifth week of sampling (Figure 5).

The susceptibility dynamics to the tested antimicrobial compounds varied across weeks (Figure 6). Several antibiotics presented significant differences among sampling weeks. For aztreonam, isolates in the first week were more susceptible than those collected in the fifth week of sampling (*p* = 0.038). Similar trends were observed for ceftazidime (*p* < 0.001). Although isolates collected in the third week were more susceptible to enrofloxacin than those from the first week (*p* = 0.017), isolates collected in the fifth week were less susceptible than those from the first week (*p* < 0.001). Regarding imipenem (*p* = 0.026), isolates collected in the fifth week were more susceptible than those collected in the first week. Regarding sulfamethoxazole/trimethoprim, isolates collected in the first week were more susceptible than those collected in the third (*p* = 0.012), the fourth (*p* = 0.004), and the fifth weeks (*p* = 0.016).

## 4. Discussion

Establishing a successful *ex situ* program is challenging. Along with the difficulties to closely recreate natural habitat conditions for the subject species in the *ex situ* settings, it is fundamental to decrease chances of disease acquisition during the program’s duration and to secure limited transmission of important pathogenic agents and their genetic information to natural habitats upon relocation in the wild. Developing husbandry techniques that help mitigate such drawbacks in an *ex situ* program is essential. In this study, we expose, for the first time, the role of aquatic *ex situ* breeding programs in the amplification of antimicrobial resistance and raise concerns regarding the subsequent introduction of antimicrobial resistance determinants into natural environments upon animal reintroductions into the wild. Similarly, we show that humans working in close association with these programs can acquire important zoonotic bacterial species when safety measures are disregarded. Our results also suggest a trend between the use of protective measures during husbandry of the animals in the *ex situ* program and the decrease of *Aeromonas* spp. prevalence and pathogenic potential.

It is important, however, to acknowledge a major limitation in this study driven by the selected experimental setup. By using only one tank to represent each of the experimental treatments evaluated, and therefore not including replicates that could evidence variability associated with each tank, this study does not fully corroborate the definitive link between the protective measures tested and the observed changes in bacterial dynamics and virulence expression. Although current findings suggest an effect of the tested measures in these parameters, other factors not controlled by the experimental set up could have influenced the dynamics in the tanks differentially. Despite the number of fish individuals available, their intrinsic characteristics, and the technical limitations present, the current study is considered an important preliminary investigation in this field and should be complemented with further investigations.

Despite the fact that *Aeromonas* spp. prevalence in fishes was similar among tanks during most of the experimental period, a decrease was observed in the last week within the test tank. Similarly, no *Aeromonas* spp. were retrieved from food sources in the test tank, whilst they were detected in the food administered to the animals in the control tank in two of the sampling procedures. It is noteworthy that the isolates collected from the food sources were clones, highlighting a common source of contamination. Members of the genus *Aeromonas* have previously been isolated from fomites [29]. In a study conducted by Bebak et al. [30], the risk of *A. hydrophila* outbreaks in cultured catfish significantly increased when animals were seined. Since disinfection protocols, although available [31], are often not regularly implemented, these type of materials represents an important fomite for pathogen introduction into fishes’ populations. Current results shed light on the role that materials commonly used during the husbandry of *I. lusitanicum* and other leuciscid species under *ex situ* conditions, such as hand nets for animal capture and plastic containers for food thawing and preparation, might have in the transmission and persistence of important aquatic pathogens in aquarium tanks used for conservational purposes. Additionally, and since *Aeromonas* spp. are commonly isolated from tap water [32,33], the use of such water to thaw frozen food for administration can constitute another transmission channel to fishes housed in captive breeding programs.

Although expected, due to the lack of disinfection protocols, it was not possible to isolate *Aeromonas* spp. from the hands and arms of the aquarists involved in this study prior to animal handling. Nevertheless, after handling *I. lusitanicum* individuals, the aquarists’ hands and arms yielded *Aeromonas* spp. The zoonotic potential of some species belonging to the *Aeromonas* genus is acknowledged and their role as emergent pathogens has been stressed in recent years [34]. By harboring a distinct cluster of *Aeromonas* spp., it seems that the most likely route for bacterial acquisition in aquarists was through contact with tank water. Additionally, some isolates collected from the hands of aquarists in distinct weeks were clones, highlighting the success that some strains have in remaining in the tank environment over prolonged periods [35]. The lack of biosafety measures such as the use of gloves or hand disinfection results in the acquisition of bacterial pathogens by the aquarists and can evolve into clinical manifestations of disease in certain situations. The development of such infections is possible in people handling aquatic species, being sometimes amplified by cuts and abrasions present in the skin, and can often result in diseases ranging from wound infection to sepsis [36,37]. Additionally, this finding highlights the possibility of aquarists acting as vectors of bacterial transmission among animals housed in different settings in captivity. Although further research would be needed to clarify this link, it is important to stress that the use of biosafety measures would likely decrease the probability of pathogen transmission during husbandry actions with aquatic species.

Although *Aeromonas* species’ structure was similar in the beginning of the trial in both tanks, shifts in the bacterial structure of both tanks were observed across the duration of the experiment. Nevertheless, an increase and predominance of *A. hydrophila* isolates was observed. This similar trend can reflect different explanations. In one hand, it is possible that a similar route of contamination existed for both tanks. As referred to before, tap water can be a vehicle for the transmission of *Aeromonas* spp. and the weekly addition of new water to the tanks could have shaped the dynamics observed. On the other hand, and since *Aeromonas* spp. structure in a certain aquatic environment is controlled by its environmental conditions [38,39,40,41,42], it is possible that conditions shared by both tanks favored the development of *A. hydrophila* in detriment of other *Aeromonas* species. Additionally, the species shifts observed for each tank possibly result from the use of protective measures in the test tank, although a definitive conclusion can’t be drawn. Actions that can be met with the introduction of new bacterial strains—i.e., the use of handling material used for individuals living in other tanks and that are not subjected to disinfection protocols—can influence the pre-existing bacterial structure by disrupting the bacterial dynamics originally experienced in the tank. Contrarily, decreasing the external pressures (i.e., by using protective measures) on a determined ecosystem that is in equilibrium will likely retain the original bacterial structure. Another important factor that ca not be downplayed are individual fish traits (i.e., genotype, mucus constitution) that can influence bacterial structure in the skin and determine bacterial colonization success. It is also important to acknowledge that, while RAPD analysis may not be considered the most suitable methodology to perform bacterial diversity assessments in *Aeromonas* spp., in the scope of this study, where clone identification and bacterial transmission routes were being evaluated, this tool has proven to be cost-effective and reliable, similarly to previous studies [10,11].

A significant reduction of the virulence expression at 22 °C was observed in the isolates from the test tank. One of the main routes for virulence acquisition in bacteria is through the integration of virulence genes originating from horizontal gene transfer, conferring a fitness advantage to the bacterial strain, along with an increased pathogenic potential [43]. Contact bridges between different hosts or environments allow for the transfer of such genetic information, along with microorganisms carrying genes absent in the original microbiota. In our study, the measures applied in the test tank appeared to prevent the exchange of such virulence determinants between used material and aquarists and *I. lusitanicum* individuals, resulting in a significantly lower virulence potential from the isolates collected in this tank. Another hypothesis is related to the bacterial communities in each tank, that differing between tanks could also present differential virulence indexes, being then independent of horizontal gene transfer. The same pattern was not observed at 37 °C. Temperature is an important driver of the expression of virulence genes, since bacterial sensory systems perceive environmental modifications and modulate their gene expression [44,45]. As a response to heat increases, *Aeromonas* spp. will up-regulate virulence pathways, which will result in the increased production of extracellular products [44,46]. As a consequence, *Aeromonas* spp. will display higher virulence and, hence, higher pathogenic potential.

High rates of multidrug resistance levels in the isolates collected in this study are alarming. Additionally, the significant increase of MAR index values in both tanks across the extent of the experiment highlight the role that breeding programs might have in the dissemination of resistance determinants to wild threatened populations after restocking actions with captive-bred animals. Although in this study we focused on aquatic animals, our results are in accordance with previous studies [5,6] conducted in *ex situ* conservation programs of terrestrial animals, reinforcing this idea. Since a similar prevalence of non-susceptibility among the isolates from both tanks were registered for the tested antimicrobial compounds, and the observed dynamics across the weeks were equal between tanks, a common source of contamination is the most likely route for the introduction of resistance determinants into the tanks. Despite the fact that different techniques are employed to remove resistance determinants, tap water still presents a source of antibiotic resistance genes [47]. Furthermore, the quantity and type of antibiotic resistance genes in the water treatment system are dynamic and dependent on the bacterial species present [48]. Hence, it is likely that the acquisition and amplification of antimicrobial resistance in both tanks, as well as the dynamics observed along the sampling weeks, are a reflection of the resistome of the water used in the tanks and are a concern for aquatic *ex situ* programs.

## 5. Conclusions

Biosafety measures should be a critical component of any *ex situ* breeding program, ensuring the successful outcome of the process. In this study, we evaluated the use of protective measures on the prevalence, structure, virulence expression, and antimicrobial susceptibility profiles of a potential fish and human pathogenic genus—*Aeromonas*. Investigations both on the use of biosafety measures, as well as on antimicrobial resistance and virulence acquisition, have lacked in breeding programs of aquatic species. Simple protective measures, such as the disinfection of the handling material and the use of gloves when manipulating animals, can be easily implemented without substantial cost increments and provide significant improvements for the animals’ welfare. Since the current study is considered a pilot approach to the problem, further research needs to be conducted in order to confirm the trends observed with the use of protective measures. Additionally, mitigation measures to decrease antimicrobial resistance acquisition, dissemination, and amplification in *ex situ* breeding programs are urged.

## Figures and Tables

**Figure 1 animals-12-00436-f001:**
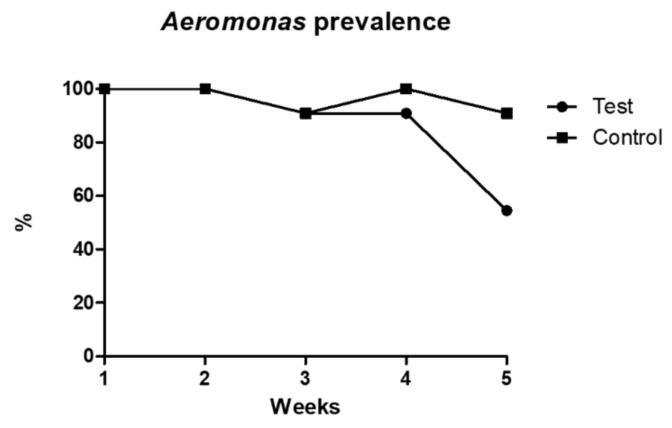
*Aeromonas* spp. prevalence in both tanks across the sampling weeks.

**Figure 2 animals-12-00436-f002:**
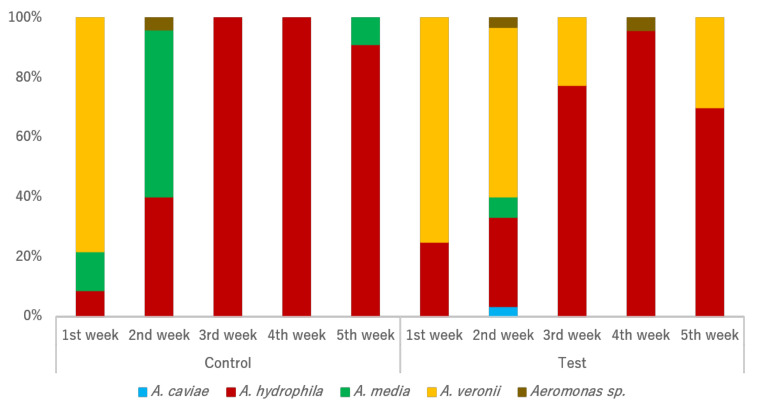
Relative prevalence of *Aeromonas* species by tank and sampling week.

**Figure 3 animals-12-00436-f003:**
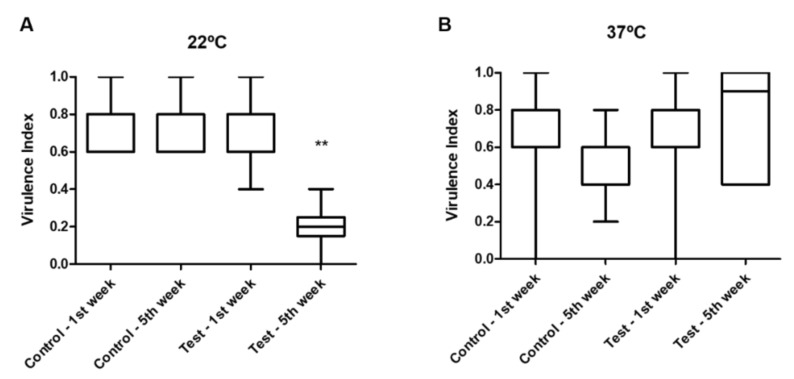
Virulence index of the isolates collected in both tanks in the 1st and 5th week and analyzed at 22 °C (**A**) and 37 °C (**B**). ** *p* < 0.01.

**Figure 4 animals-12-00436-f004:**
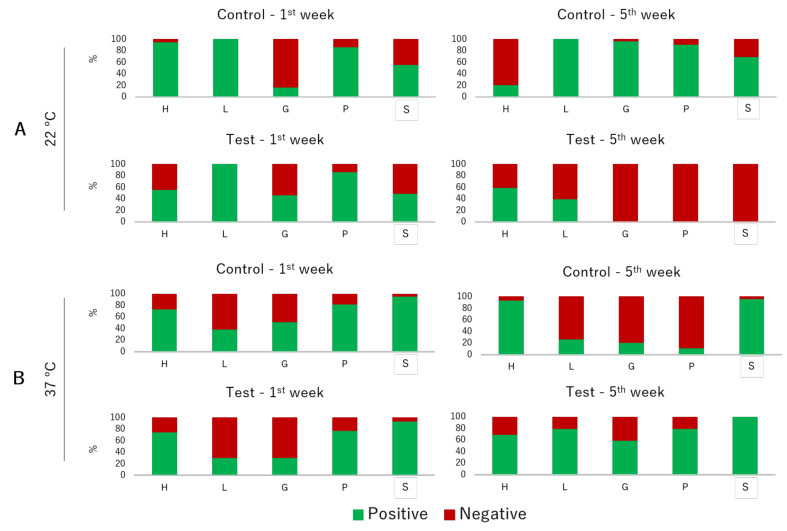
Relative prevalence of virulence factors by tank and sampling week of the isolates analyzed at 22 °C (**A**) and 37 °C (**B**). H—hemolytic activity, L—lipase activity, G—gelatinase activity, P—protease activity, S—slime production.

**Figure 5 animals-12-00436-f005:**
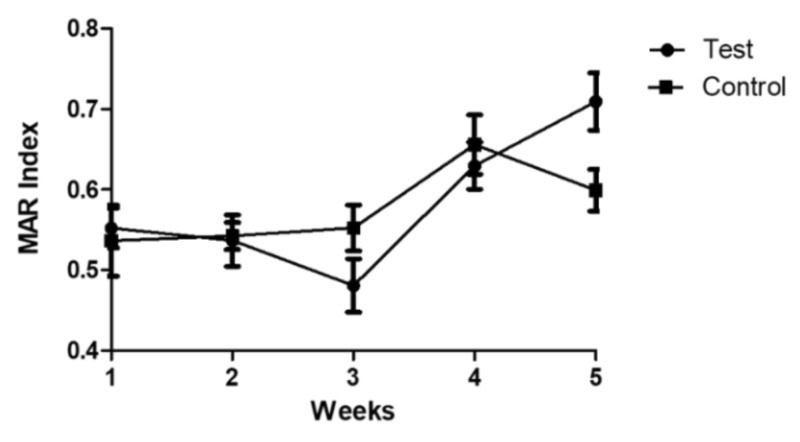
MAR index values (mean + SEM) of the isolates collected in both tanks across the sampling weeks.

**Figure 6 animals-12-00436-f006:**
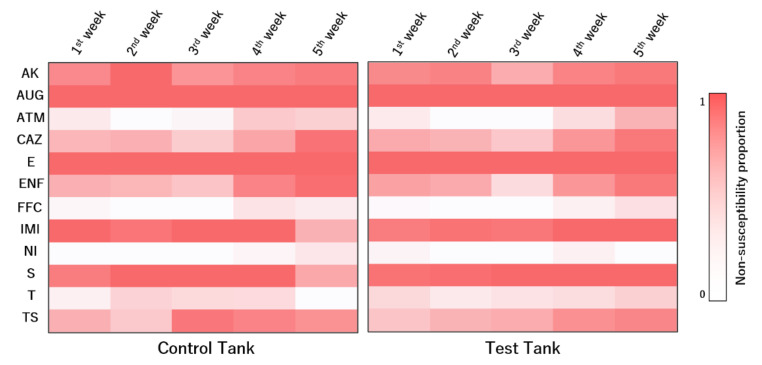
Relative prevalence of non-susceptibility to the tested antimicrobial compounds of the isolates collected in both tanks across the sampling weeks. AK—amikacin, AUG—amoxicillin/clavulanic acid, ATM—aztreonam, CAZ—ceftazidime, E—erythromycin, ENF—enrofloxacin, FFC—florfenicol, IMI—imipenem, NI—nitrofurantoin, S—streptomycin, T—tetracycline, TS—sulfamethoxazole/trimethoprim.

**Table 1 animals-12-00436-t001:** Experimental conditions in this study.

Action	Control Tank	Test Tank
Tank covering	Fine mesh	Plastic cover
Food preparation	Food items thawed with tap water in non-disinfected containers	Food items thawed with sterile water transported in sterile shots in containers disinfected with 70% ethanol and exposed to UV radiation
Water renovation and food surplus retrieval	Water and food pumped out of the tank with a non-disinfected suction system	Water and food pumped out of the tank with a suction system disinfected with 70% ethanol and exposed to UV radiation
Fish handling	Operator not using gloves and using non-disinfected shrimp nets and handling tanks	Operator using nitrile gloves and handling tanks disinfected with 70% ethanol and shrimp nets disinfected with 70% ethanol and exposed to UV radiation
Fish sampling	Measuring device non-disinfected	Measuring device disinfected with 70% ethanol

## Data Availability

The data resulting from this study is available upon request to authors.

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
