# Peer review of "Aeromonas spp. Prevalence, Virulence, and Antimicrobial Resistance in an Ex Situ Program for Threatened Freshwater Fish—A Pilot Study with Protective Measures"

_animals, 2022, doi:10.3390/ani12040436_

Round 1
Reviewer 1 Report
In this short communication, which evaluated the ex situ breeding program in prevalence, virulence, and antimicrobial resistance of Aeromonas in critically endangered Iberochondrostoma lusitanicum fish. The manuscript has an importance in the field of RAS and conservation studies of the endangered species. The MS was written in a good language. It could be considered after minor corrections.
- In the experimental design: they used one replicate for each group contains 11 fish. This is low number of replicates to have reliable results.
- Did this study have ethical approval for handling, sampling, and especially fish collection by electrofishing.
- Could you use the common name of Iberochondrostoma lusitanicum with scientific name at the first mention in the abstract and introduction, then use the common name or abbreviated Latin name for subsequent use.
- L 86 : italicise the Latin name and consider along the manuscript “Iberochondrostoma lusitanicum”. L 95, L 103, L 106: Aeromonas spp.
- L 88: italicise the word “ex situ” along the MS.
- L 119-120: rewrite this sentence.
- Figure 1: add stars for the weeks that have significant differences.
- Figure 2: could you use high resolution (600 dpi).
Author Response
Dear Dr. Kelsey Li
We thank the reviewers for their thoughtful comments and helpful critique that have contributed to improve the manuscript. Please find below a point by point reply to the reviewers where we carefully address their concerns and inputs.
On behalf of the authors,
Miguel Grilo

Reviewer 2 Report
This manuscript deals with the ex situ conservation program of the aquatic species with understanding of prevalence and antimicrobial resistance and virulence determinants of Aeromonas spp., an important zoonotic and fish pathogenic agent.
As overall, the methods and results are well presented, including images which help readers to understand the procedures. However, some points (esp on experimental setup, water quality data) in the comments attach.
For the discussion, it can be improved. The author should add why procedure is unique, what are the major findings, are the results are strong or valid? And also discuss the relevance to current literature and should correlate her data/analysis with the results from previous studies. Final paragraph ; Significant of the study–Add the info on the study will help the authorities to control the transmission of bacterial pathogens in case of potential Aeromonas infection or outbreaks.
I have pointed some points in the comments in the attach. I think the manuscript should be revised for minor errors (see attach) before publication.

Author Response

(The authors gave the same response as above.)

Reviewer 3 Report
The research question is interesting. The whole manuscript describes a pilot study aimed to investigate the role of protective measures in the prevalence, antimicrobial resistance profiles and virulence signatures of Aeromonas spp. in Iberochondrostoma lusitanicum kept in an ex situ program. In my opinion, it is necessary to perform studies to evaluate the dissemination of pathogenic bacteria by fishes, as the authors state in the Conclusion: “Biosafety measures should be a critical component of any ex situ breeding program”.
However, I do not think the article is ready to be published. The manuscript has many flaws and gaps. In addition, it is too summarized and some important information are lacking, mainly in the methodology. I could understand a little bit some procedures only after reading the Appendix A. The authors have to provide more complete information in the Methodology of the article. Some methods they could write from the Appendix text. Table 1 is also confusing and it was not adequately prepared. Furthermore, there are some incomplete sentences from line 117 to 124 (for example: “For each isolate, the virulence [17]”???). The authors need to review all Materials and Methods and prepare a more comprehensive text for the readers. It is also not clear how the authors evaluate the virulence. It would be important to separate the analysis of the virulence from the antimicrobial resistance.
Results were also not properly evaluated and presented. The first three paragraphs seem Ok. However the 4th paragraph describes 58 clones with 91.88% of similarity. The authors also provide four Supplemental figures, but they do not cite them in the text. They also present results from different Aeromonas species together. They should present the results of all species separately. RAPD is a molecular biology methodology with severe limitations to study bacterial diversity. So I would suggest the authors to evaluate the diversity for each Aeromonas species separately. Finally, the Discussion is poor and the authors have not compared presenting could approach other important articles and methodologies to evaluate bacterial diversity (suchs as MLST and WGS).
Therefore my main suggestion to the authors is to revise the whole manuscript and prepare a version for resubmission.
Author Response

(The authors gave the same response as above.)

Round 2
Reviewer 3 Report
The manuscript was improved. The authors reviewed the Materials and Methods and prepared a more comprehensive text for the readers. However they have basically included all the information from the Appendix A in the Methodology. Material & Methods are too extensive now (it is almost half of the manuscript!) with many unnecessary information. Table 2 and Figure 1 could be supplemental material, for example. In addition, all text from this section could be summarized. I suggest the authors an additional effort to review again this section. Finally, the authors could also insert the argumentation to use RAPD in the Discussion, as they answered in the rebuttal letter: "it is important to state that the goal in this study
regarding the use of molecular typing was mostly to signal bacterial clones,
evaluate possible colonization routes between fishes, food, water and aquarists and to eliminate clones from subsequent analysis (i.e., virulence and multiple antibiotic resistance indexes. We chose to use RAPD analysis due to its simplicity, reduced cost and proven applicability with Aeromonas collections (Szczuka & Kaznowski, 2004; https://doi.org/10.1128/JCM.42.1.220-228.2004).) "
Author Response

(The authors gave the same response as above.)
